# Highly Enhanced Electrocatalytic Performances with Dendritic Bimetallic Palladium-Based Nanocrystals

**Respati K. Pramadewandaru [1]**, **Jeong-Hu Shim [1]**, **Young Wook Lee [2],***  and **Jong Wook Hong [1],***

[1]  Department of Chemistry, University of Ulsan, Ulsan 44776, Korea; respatikevin@gmail.com (R.K.P.); wjdgn3524@gmail.com (J.-H.S.)

[2]  Department of Chemistry Education and Research Institute of Natural Sciences, Gyeongsang National University, Jinju 52828, Korea

*  Correspondence: lyw2020@gnu.ac.kr (Y.W.L.); jwhong@ulsan.ac.kr (J.W.H.); Tel.: +82-55-772-2227 (Y.W.L.); +82-52-712-8013 (J.W.H.)

**Abstract:** The exploration of efficient nanocatalysts with high activity and stability towards water electrolysis and fuel cell applications is extremely important for the advancement of electrochemical reactions. However, it remains challenging. Controlling the morphology of bimetallic Pd–Pt nanostructures can be a great way to improve their electrocatalytic properties compared with previously developed catalysts. Herein, we synthesize bimetallic Pd–Pt nanodendrites, which consist of a dense matrix of unsaturated coordination atoms and high porosity. The concentration of cetyltrimethylammonium chloride was significant for the morphology and size of the Pd–Pt nanodendrites. Pd–Pt nanodendrites prepared by cetyltrimethylammonium chloride (200 mM) showed higher activities towards both the hydrogen evolution reaction and methanol oxidation reaction compared to their different Pd–Pt nanodendrite counterparts, commercial Pd, and Pt catalysts, which was attributed to numerous unsaturated surface atoms in well-developed single branches.

**Keywords:** nanodendrites; alloy nanocrystals; Pd–Pt; electrocatalysis; hydrogen evolution reactions; methanol oxidation reaction





## 1. Introduction

The preparation of metal nanocrystals (NCs) is significant for the development of efficient catalysts for various promising electrochemical reactions [1,2]. In particular, controlling the shape, size, and composition of metal NCs can allow the interaction of target molecules with the surface of NCs to be optimized, resulting in the enhancement of catalytic performances [3–5]. Recently, porous NCs have attracted an enormous amount of interest due to their enhanced catalytic performances, attributable to their large surface areas [6,7]. Among various porous metal NCs, dendritic NCs with many highly porous branches have attracted much attention for their morphological benefits, such as high volume-to-surface area, efficient mass transport, and many unsaturated surface atoms in their branches, which can promote the conversion of molecules for various electrochemical reactions [8–11].

Along with the shape of noble metal NCs, controlling the composition of the NCs is effective in enhancing their catalytic performance. The incorporation of secondary metals into monometallic NCs can influence the adsorption strength of reagents and form an intermediate by changing the electronic structures of active surface metal atoms [12–16]. In addition, some bimetallic NCs can remove poisoning intermediates bound on active surface atoms, and bimetallic NCs with controlled compositions have shown enhanced electrocatalytic performance compared with monometallic NCs [17–19]. In particular, Pd–Pt bimetallic NCs constructed by precise shape and composition engineering have exhibited remarkable electrocatalytic activities towards various electrochemical reactions, including the methanol oxidation reaction (MOR) and hydrogen evolution reaction (HER) [20–24].

Based on these previous findings, it can be anticipated that using Pd–Pt bimetallic nanodendrites (NDs) with a small size is a desirable approach for the development of efficient electrocatalysts for various electrochemical reactions.

In this work, we report wet chemical synthesis for the preparation of Pd–Pt bimetallic NDs with controllable sizes. Pd–Pt alloy NDs were produced by the co-reduction of $Na_2PdCl_4$ and $K_2PtCl_4$ in the presence of cetyltrimethylammonium chloride (CTAC) and ascorbic acid (AA), used as stabilizers and reducing agents, respectively. CTAC and AA are critical in creating the well-defined dendritic shape and compositional structure of the bimetallic Pd–Pt composition. Notably, adjusting the amounts of CTAC in the synthesis allowed precise manipulation of the size of the Pd–Pt NDs, resulting in various Pd–Pt NDs with different sizes. The Pd–Pt NDs with different sizes exhibited distinctive electrocatalytic performances against both MOR and HER. For MOR and HER, the Pd–Pt NDs with an average diameter size of 29.2 ± 4.9 nm, prepared using 200 mM of CTAC, exhibit higher catalytic activity compared to their Pd–Pt NDs counterparts with different sizes, commercial Pd/C, and Pt/C catalysts, due to their highly porous morphology, favorable exposed facet, and Pd–Pt bimetallic properties.

## 2. Results and Discussion

Figure 1a,b shows the representative TEM images of the product prepared by the co-reduction of $Na_2PdCl_4$ and $K_2PtCl_4$ by AA in the presence of CTAC (200 mM), demonstrating that the majority of the nanostructures consisted of NDs with highly porous branches. The NDs possess an average diameter size of 29.2 ± 4.9 nm and branch thickness size of 8.7 ± 0.9 nm. The high-resolution TEM (HR-TEM) image of an ND displays d-spacing of 2.24 Å between the adjacent lattices. This highly corresponds to that of the (111) planes of face-centered cubic (fcc) bimetallic Pd–Pt (inset in Figure 1b) [25]. The fast Fourier transform (FFT) pattern obtained from an ND reveals the highly crystalline feature of the prepared NCs (inset of Figure 1b). Notably, as shown in the inset in Figure 1b, many unsaturated surface atoms at the tips of the ND were explicitly observed in the HR-TEM image. To further investigate the structural characteristics of the NDs, high-angle annular dark-field scanning TEM (HAADF-STEM) images of NDs were obtained (Figure 1c).

The high porosity of the NDs was observed by a strong contrast in the NDs. The formation of a bimetallic Pd–Pt core–shell structure was corroborated by elemental mapping and overlap profile analysis with HAADF-STEM energy-dispersive X-ray spectroscopy (HAADF-STEM-EDS) (Figure 1c,d). The Pd/Pt atomic ratio of the NDs, measured by EDS analysis and ICP-OES, were 51:49 and 53:47, respectively. In addition, the diffraction peaks in the XRD pattern of the NDs further show their Pd–Pt bimetallic feature (Figure 1e) [21,26,27].

The NDs with different diameter sizes and porosity were realized by changing the amount of CTAC added into the reaction mixture. The representative SEM and TEM images of the products show that the average diameter sizes of the NDs increased as the amount of CTAC increased (Figure 2). The CTAC solution (200 mM) resulted in the formation of Pd–Pt–$200_{CTAC}$ NDs with an average diameter size of 29.2 ± 4.9 nm (standard synthesis). Pd–Pt NDs with average sizes of 46.3 ± 7.7, 38.6 ± 8.9, and 36.5 ± 5.6 nm (Pd–Pt–$10_{CTAC}$, Pd–Pt–$50_{CTAC}$, and Pd–Pt–$100_{CTAC}$ NDs, respectively) were formed when different concentrations of CTAC solutions (10, 50, and 100 mM) were added for the synthesis of Pd–Pt nanostructures (Figure 2a–c). Along with the change in size, NDs obtained by a higher concentration of CTAC show a high degree of porosity due to their more developed branches, indicating higher porosity of Pd–Pt–$200_{CTAC}$ NDs than the other NDs (inset of Figure 2e–h).

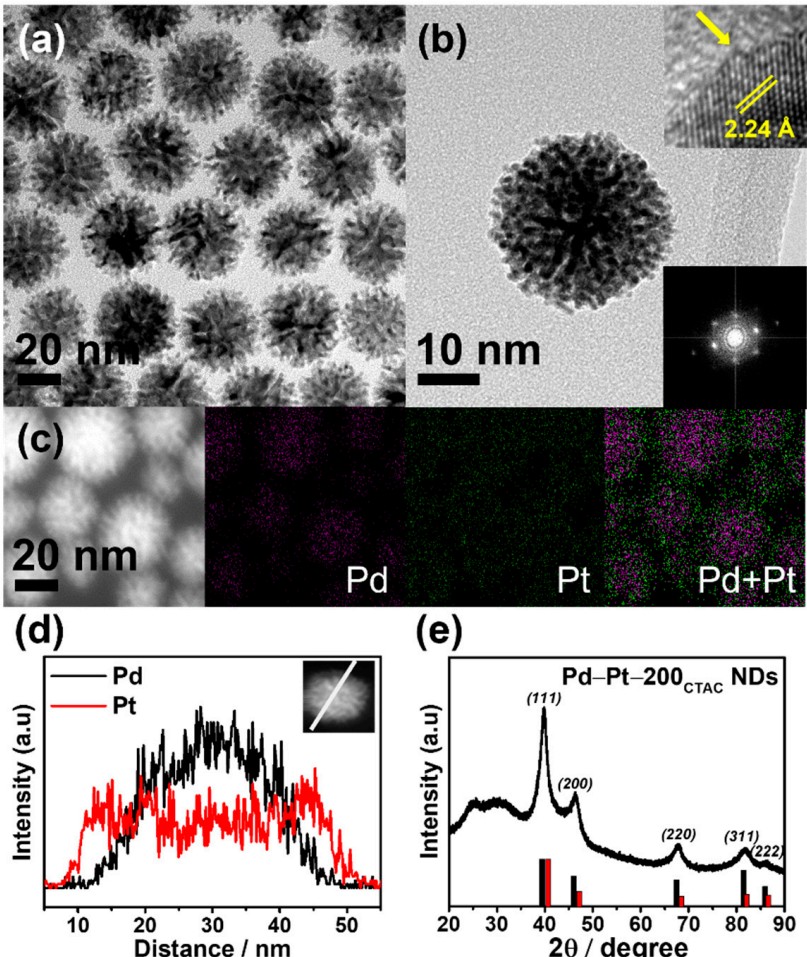

**Figure 1.** (**a**,**b**) HR–TEM image including the correspond FFT images. HAADF-STEM image with (**c**) corresponding EDS elemental mapping images, and (**d**) line-scan profile analysis of Pd (black) and Pt (red) for the Pd–Pt NDs. (**e**) XRD pattern confirm of corroborating the Pd–Pt bimetallic feature.

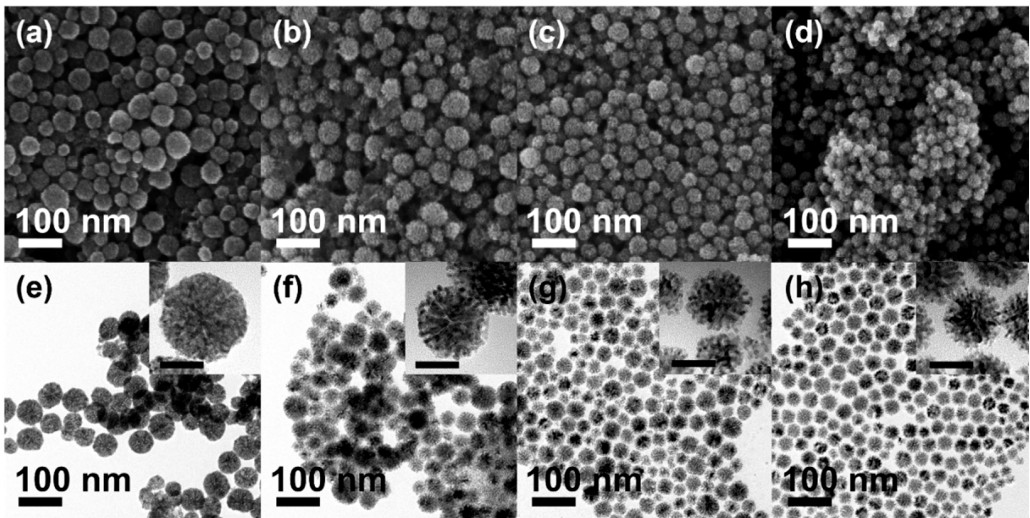

**Figure 2.** SEM and TEM images of Pd–Pt NDs with controlled CTAC concentration representative to (**a**); (**e**) Pd–Pt–10$_{CTAC}$ NDs, (**b**); (**f**) Pd–Pt–50$_{CTAC}$ NDs, (**c**); (**g**) Pd–Pt–100$_{CTAC}$ NDs, and (**d**); (**h**) Pd–Pt–200$_{CTAC}$ NDs. Scale bar in the inset figures of parts e–h is 30 nm.

The XRD patterns of the samples also show the bimetallic Pd–Pt nature, which is analogous with that of Pd–Pt–200$_{CTAC}$ NDs (Figure 3). In addition, similar peak positions in the XRD patterns for various Pd–Pt NDs were observed, which reveals the analogous composition structures.

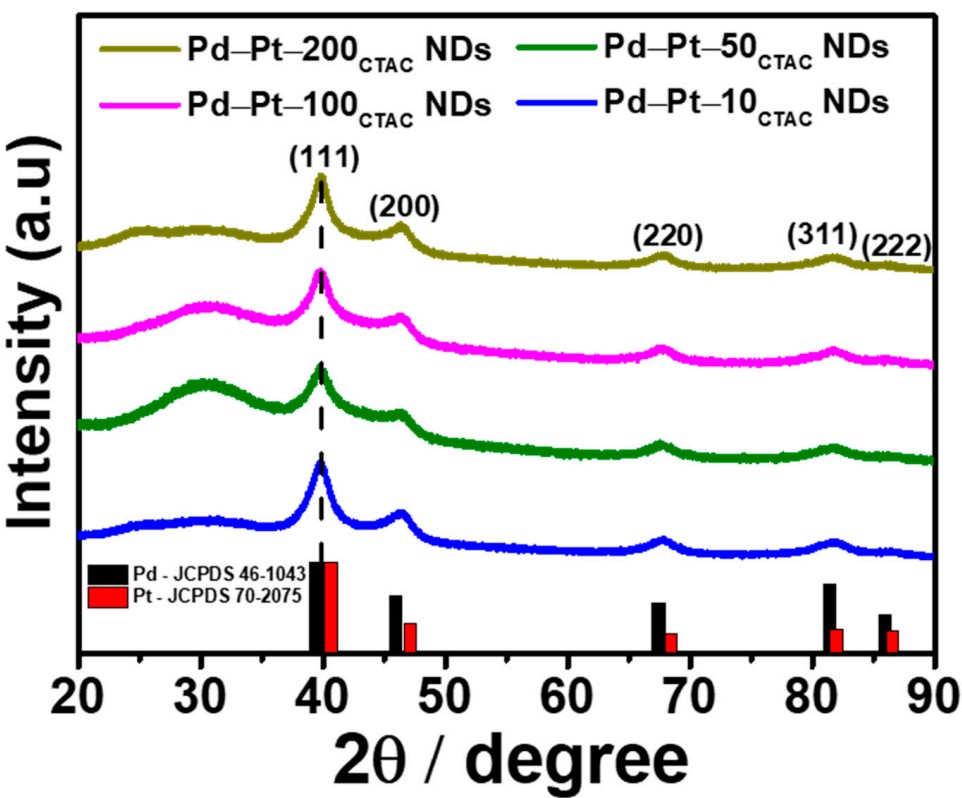

**Figure 3.** XRD pattern of Pd–Pt NDs with controlled CTAC concentrations. The line pattern shows reference cards 46-1043 and 70-2075 for Pd and Pt according to JCPDS.

The different sizes of NDs prepared by different amounts of CTAC indicate that CTAC and metal precursors closely interplay, resulting in different sizes and porosity.

To further investigate the influence of CTAC concentration on the formation of NDs, nanostructures were produced in reaction mixtures with various concentrations of CTAC, while other synthesis conditions were unchanged. In a higher concentration of CTAC (500 mM), products with a similar shape to the Pd–Pt–200$_{CTAC}$ NDs were synthesized (Figure 4a), although their size is somewhat larger (48.3 ± 3.4 nm). In contrast, a lower concentration than 10 mM of CTAC, such as 2.5, 5.0, and 7.5 mM, yielded some irregular shaped nanostructures (Figure 4b–d). In a low concentration of CTAC, although large amounts of seed NCs can be initially produced, insufficient surfactants adsorbed on the surface of seed NCs can lead to fast aggregation of pre-formed seed NCs, resulting in the formation of NDs with uncontrolled shapes and large diameter sizes, which further demonstrates the importance of an optimal CTAC concentration for the formation of dendritic shapes [10,23,28]. Given that NDs with well-developed branches and smaller sizes possess a larger volume-to-surface area than NDs with unmatured branches and larger sizes, we expect that the Pd–Pt–200$_{CTAC}$ NDs can exhibit improved electrocatalytic performances.

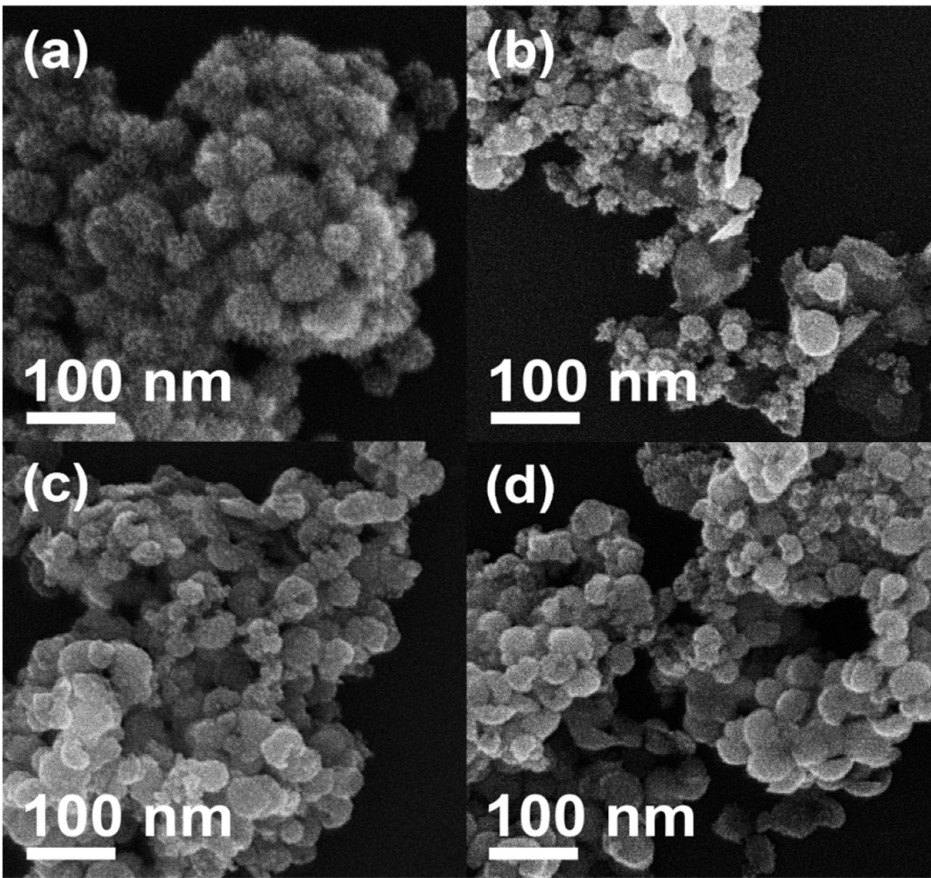

**Figure 4.** SEM images of Pd–Pt NDs prepared with controlled concentrations of CTAC with higher concentration of (**a**) 500 mM. Pd–Pt NDs prepared with lower concentration of CTAC than 10 mM with (**b**) 2.5 mM, (**c**) 5.0 mM and (**d**) 7.5 mM.

To investigate the effect of dendritic structure and size on the bimetallic Pd–Pt NDs, the electrocatalytic activities of the various Pd–Pt NDs (Pd–Pt–200$_{CTAC}$, Pd–Pt–100$_{CTAC}$, Pd–Pt–50$_{CTAC}$, and Pd–Pt–10$_{CTAC}$ NDs) were estimated for HER, and their electrocatalytic activities were compared to those of both commercial Pd/C and Pt/C catalysts (Figure 5). The CVs of various catalysts were measured in $N_2$-saturated 0.5 M $H_2SO_4$ at a sweeping rate of 50 mV s$^{-1}$ (Figure 5a) to remove residual surfactants and organic molecules adsorbed onto the surface of the catalysts [29]. The HER performances were examined by LSV polarization curves at a sweeping rate of 10 mV s$^{-1}$ within the applied potential window of 0.05 to −0.25 V vs. RHE at room temperature (Figure 5b,c). According to the LSV curves, the overpotentials of the Pd–Pt–200$_{CTAC}$ NDs, Pd–Pt–100$_{CTAC}$ NDs, Pd–Pt–50$_{CTAC}$ NDs, Pd–Pt–10$_{CTAC}$ NDs, and Pd/C and Pt/C catalysts at the current density (10 mA cm$^{-2}$) reach 32.3, 32.4, 33.1, 34.7, 122.3, and 35.4 mV, respectively. To further evaluate the catalytic activities of the catalysts, the linier sections of the Tafel plots shown in Figure 5b,c were fitted to the Tafel equation, and the slope values were obtained (Figure 5d) [30]. The Tafel slope values of the Pd–Pt–200$_{CTAC}$, Pd–Pt–100$_{CTAC}$, Pd–Pt–50$_{CTAC}$, and Pd–Pt–10$_{CTAC}$ NDs were 23.8, 24.4, 24.7, and 25.1 mV dec$^{-1}$, respectively, which are lower than those of both commercial Pd/C (76.5 mV dec$^{-1}$) and Pt/C (30.5 mV dec$^{-1}$) catalysts (Table 1). The findings signify the superior HER activities of all bimetallic Pd–Pt NDs compared to commercial Pd/C and Pt/C catalysts. Noticeably, the Pd–Pt–200$_{CTAC}$ NDs exhibited higher electrochemical HER activity than different Pd–Pt NDs, which implies the importance of dendritic shapes with well-developed branches.

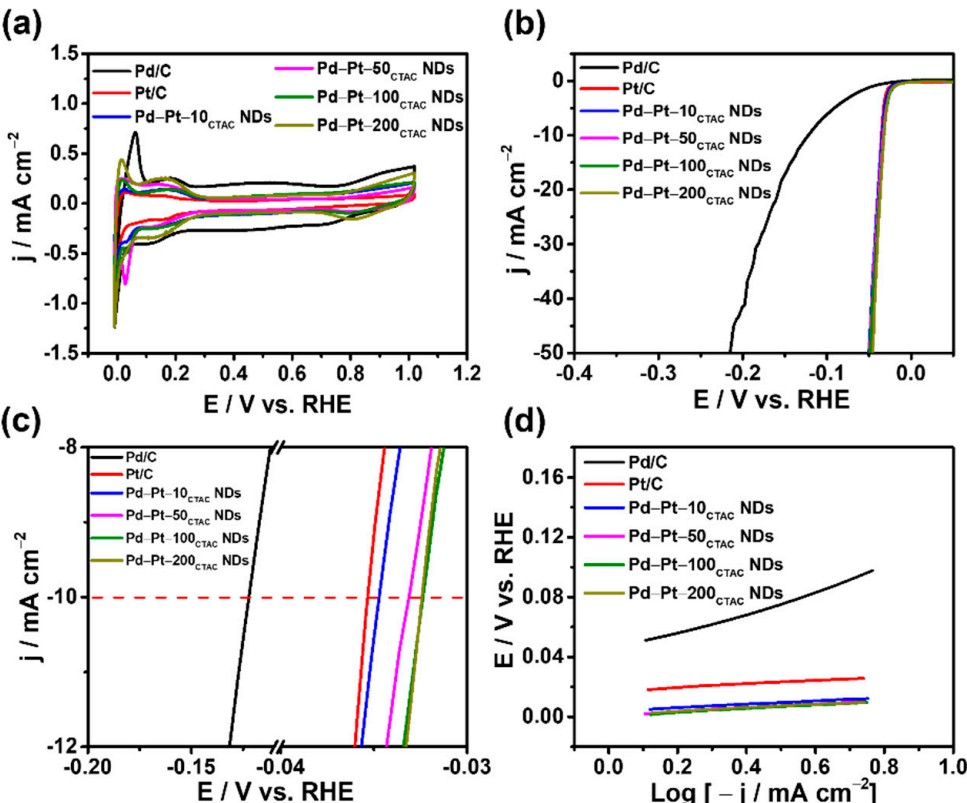

**Figure 5.** Electrocatalytic HER performance of different Pd–Pt NDs, and commercial Pd/C and Pt/C electrocatalysts measured in 0.5 M $H_2SO_4$ saturated with $N_2$; sweep rate at 50 mV s$^{-1}$ (CVs) and 10 mV s$^{-1}$ (LSV). (**a**) CVs of all Pd–Pt electrocatalysts. (**b**) LSV polarization curves and (**c**) derived from (**b**) to see details of $H_2$ evolution in overpotential at 10 mA cm$^{-2}$ of all electrocatalysts including commercials. (**d**) Corresponding Tafel plots recorded on the electrode by polarization curves at potential 10 mV s$^{-1}$.

**Table 1.** Activity values for all electrocatalysts including commercial Pd/C and Pt/C in HER activity. The values including overpotential at 10 mA cm$^{-2}$ and Tafel plots compared in 0.5 M $H_2SO_4$.

| Catalyst | Overpotential/(mV) ~10 mA cm$^{-2}$ | Tafel Plots/(mV dec$^{-1}$) |
|---|---|---|
| Pd/C | 122.3 mV | 76.8 mV dec$^{-1}$ |
| Pt/C | 35.3 mV | 30.5 mV dec$^{-1}$ |
| Pd–Pt–10$_{CTAC}$ NDs | 34.7 mV | 25.1 mV dec$^{-1}$ |
| Pd–Pt–50$_{CTAC}$ NDs | 33.1 mV | 24.7 mV dec$^{-1}$ |
| Pd–Pt–100$_{CTAC}$ NDs | 32.4 mV | 24.4 mV dec$^{-1}$ |
| Pd–Pt–200$_{CTAC}$ NDs | 32.3 mV | 23.8 mV dec$^{-1}$ |

To further investigate the electrocatalytic properties of bimetallic Pd–Pt–200$_{CTAC}$ NDs, an electrochemical MOR was performed with various catalysts, including different NDs, commercial Pd/C, and Pt/C. The CVs of various catalysts were obtained in $N_2$-saturated 1.0 M KOH (Figure 6a).

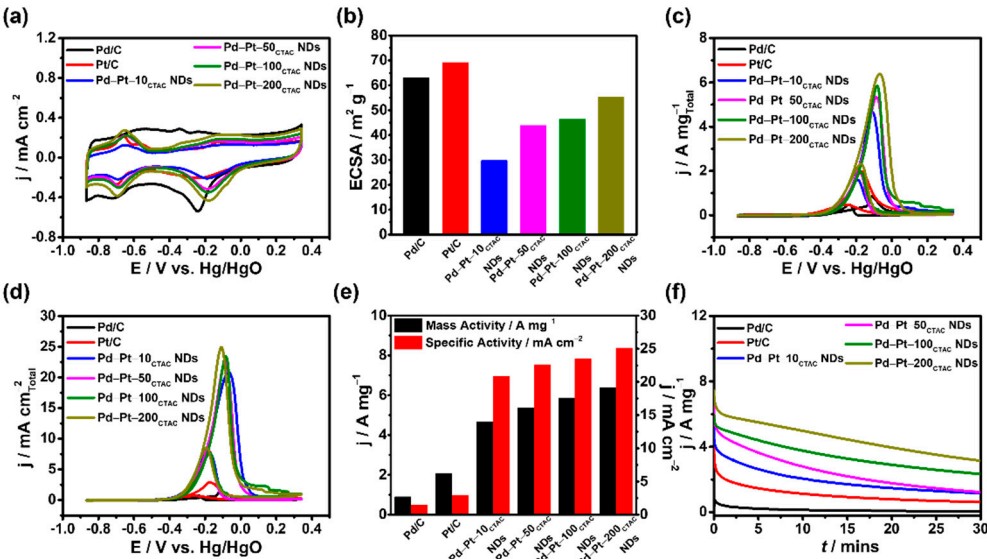

**Figure 6.** MOR performance of different Pd–Pt$_x$ NDs and commercial Pd/C and Pt/C electrocatalysts measured in 1.0 M KOH containing 1.0 M methanol saturated with N$_2$; sweep rate at 50 mV s$^{-1}$. (**a**) CVs of all catalysts and (**b**) ECSA of all values of different electrocatalysts include commercials. MOR of all metals (Pd + Pt) (**c**) mass-normalized CV curves, (**d**) specific activity-normalized CV curves, and (**e**) mass and specific activity data based on all different electrocatalysts. (**f**) Durability evaluation by chronoamperometry tests at −0.15 V vs. Hg/HgO for 30 min of all electrocatalysts.

Electrochemically active surface areas (ECSAs) of the catalysts were measured using the reduction charge of oxygen species in the CVs of the catalysts. The ECSAs for the Pd–Pt–200$_{CTAC}$ NDs, Pd–Pt–100 $_{CTAC}$ NDs, Pd–Pt–50$_{CTAC}$ NDs, Pd–Pt–10$_{CTAC}$ NDs, and Pd/C and Pt/C catalysts were 55.4, 46.4, 43.9, 29.8, 63.1, and 69.2 m$^2$ g$^{-1}$, respectively (Figure 6b), which shows that the higher ECSA value of Pd–Pt–200$_{CTAC}$ NDs than the other Pd–Pt NDs can be attributed to their smaller size and well-developed branches, leading to high porosity. To check the morphological benefits of the Pd–Pt–200$_{CTAC}$ NDs for electrocatalytic reactions, the CVs of the catalysts in 1.0 M KOH containing 1.0 M methanol were measured, and their mass activities were obtained by normalizing the current densities with the total mass (Pd + Pt) of the catalysts loaded onto the glassy carbon electrode (GCE) used as a working electrode (Figure 6c). The mass activities of the Pd–Pt–200$_{CTAC}$, Pd–Pt–100$_{CTAC}$, Pd–Pt–50$_{CTAC}$, and Pd–Pt–10$_{CTAC}$ NDs for MOR were 6.40, 5.87, 5.37, and 4.67 A mg$_{Pt}$$^{-1}$, respectively. Notably, all the bimetallic Pd–Pt NDs showed better MOR activities than both commercial Pd/C (0.90 A mg$_{Pd}$$^{-1}$) and Pt/C (2.06 A mg$_{Pt}$$^{-1}$) catalysts (Figure 6c,e), which can be attributed to the undercoordinated surface atoms formed by their unique dendritic shapes and bimetallic Pd–Pt surface. In previous reports, undercoordinated surface atoms were shown to promote the conversion of reagents to products by optimizing the binding strength between the surface of nanostructures and intermediates, which could endow them with greatly enhanced electrocatalytic activity [8]. In addition, compared with pure Pd and Pt catalysts, Pd–Pt bimetallic catalysts could reduce the adsorption of surface poisoning intermediates due to changes in the electronic structures of their surface atoms, boosting the electrochemical reactions. On the other hand, among the different bimetallic Pd–Pt NDs, the Pd–Pt–200$_{CTAC}$ NDs exhibited the largest mass activity among the different NDs for MOR; the Pd–Pt–200$_{CTAC}$ NDs showed 1.09-, 1.19-, and 1.37-times higher mass activity than Pd–Pt–100$_{CTAC}$, Pd–Pt–50$_{CTAC}$, and Pd–Pt–10$_{CTAC}$ NDs, respectively.

Furthermore, the corresponding current density of the Pd–Pt–200$_{CTAC}$ NDs, which is calculated by normalizing the current density with the ECSAs of the catalysts, is 25.12 mA cm$^{-2}$, which is 1.06-, 1.11-, 1.20-, 17.56-, and 8.43-fold larger than those of the Pd–Pt–100$_{CTAC}$ NDs (23.49mA cm$^{-2}$), Pd–Pt–50$_{CTAC}$ NDs (22.64 mA cm$^{-2}$), Pd–Pt–10$_{CTAC}$ NDs (20.89 mA cm$^{-2}$), and Pd/C (1.43 mA cm$^{-2}$) and Pt/C (2.98 mA cm$^{-2}$) catalysts, re-

spectively (Figure 6d,e). The mass and specific activities for the MOR of various bimetallic Pd–Pt NDs show a similar trend across the different sizes of Pd–Pt NDs against the counterparts of commercial Pd/C and Pt/C. Taken together, these electrochemical experiments demonstrate the synergistic influence of the dendritic shape and Pd–Pt bimetallic feature on increasing electrocatalytic performances [31–33]. The MOR activity of the Pd–Pt–200$_{CTAC}$ NDs was compared with those of previously reported Pd–Pt-based catalysts (Table 2). The Pd–Pt–200$_{CTAC}$ NDs showed superior activities compared to the previously reported catalysts, which could be due to the dendritic shape and bimetallic Pd–Pt composition structure [27,34–42].

**Table 2.** Comparative results for electrochemical MOR activity based on previous reports using Pd–Pt-based catalysts in the literature.

| Catalyst | Electrochemical Condition | Mass Activity (A mg$^{-1}$$_{total}$) | Ref. |
| --- | --- | --- | --- |
| Pd–Pt–200$_{CTAC}$ Nanodendrites | 1 M KOH + 1 M Methanol | 6.40 | **In this work** |
| Pd$_{45}$Pt$_{55}$ Nanowires | 1 M KOH + 1 M Methanol | ~1.90 | [34] |
| h-BN/PdPt Nanocorals | 1 M KOH + 0.5 M Methanol | 0.96 | [35] |
| PtPd/RGO nanogarlands | 1 M KOH + 1 M Methanol | 0.33 | [36] |
| Pt$_{50}$Pd$_{50}$ Nanocubes | 1 M KOH + 1 M Methanol | 0.34 | [37] |
| PdPt/CNTs | 0.5 M KOH + 0.5 M Methanol | 1.07 | [38] |
| Pt$_{30}$Pd$_{70}$/C | 1 M KOH + 1 M Methanol | 0.72 | [39] |
| *o*-PdH0.43@Pt Nanooctahedra | 1 M KOH + 1 M Methanol | 3.68 | [27] |
| *c*-PdH0.43@Pt Nanocubes | 1 M KOH + 1 M Methanol | 2.14 | [27] |
| PtPd Nanowires | 1 M KOH + 1M Methanol | 4.29 | [40] |
| PdPt Bimetallic Nanosponges | 1 M KOH + 1 M Methanol | ~2.20 | [41] |
| Pt@Pd/RGO | 1 M KOH + 1 M Methanol | ~0.65 | [42] |

To investigate the stability of the prepared catalysts, chronoamperometric (CA) curves of the various Pd–Pt NDs, and Pd/C and Pt/C catalysts were obtained at −0.15 V vs. Hg/HgO in a solution containing 1.0 M KOH + 1.0 M methanol for 30 min (Figure 6f). During the stability test, both commercial Pd/C and Pt/C catalysts revealed a fast decay in the current density. In contrast, the current density of Pd–Pt NDs in MOR was well maintained compared to those of commercial Pd/C and Pt/C catalysts after 30 min. In particular, the Pd–Pt–200$_{CTAC}$ NDs showed the best stability among the various Pd–Pt NDs. Furthermore, the CVs of the Pd–Pt–200$_{CTAC}$ NDs, and Pd/C and Pt/C catalysts were obtained after CA measurement (30 min) to check the superior stability of the Pd–Pt–200$_{CTAC}$ NDs (Figure 7). After the operation of CA for 30 min, the mass activity of the Pd–Pt–200$_{CTAC}$ NDs was determined as 6.1 A mg$_{Pd+Pt}$$^{-1}$, which is a 4.7% decrease compared with the initial mass activity of the Pd–Pt–200$_{CTAC}$ NDs, whereas the Pd/C (0.7 A mg$_{Pd}$$^{-1}$) and Pt/C (1.7 A mg$_{Pt}$$^{-1}$) catalysts exhibited 22.2% and 14.3% decreases compared to those obtained before the operation of CA for 30 min, respectively.

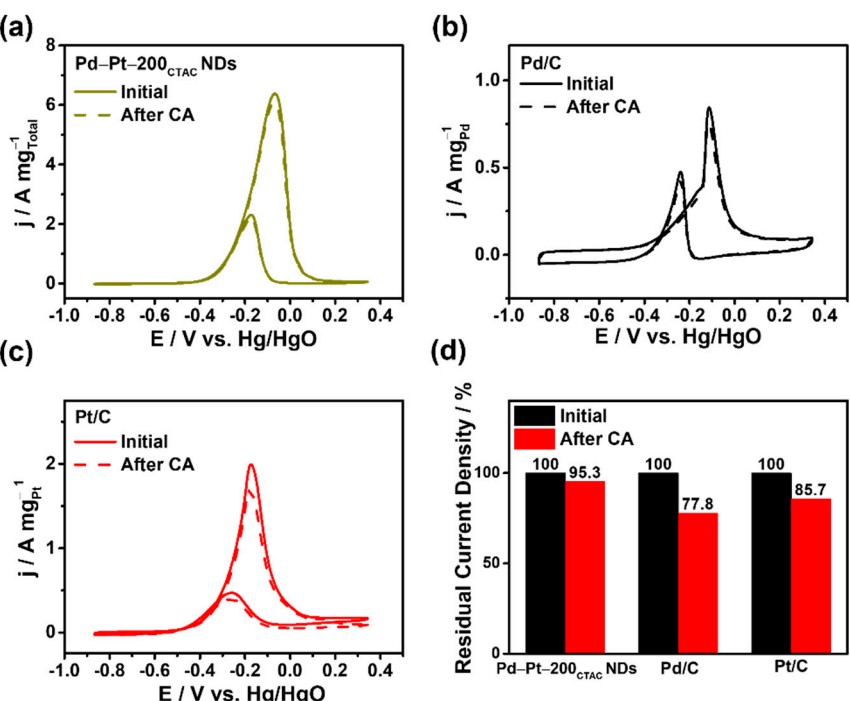

**Figure 7.** MOR residual current density activities of (**a**) Pd–Pt–200$_{CTAC}$ NDs, (**b**) Pd/C and (**c**) Pt/C after chronoamperometry experiment for 30 min. (**d**) The percentage residual current density before and after chronoamperometry measurement.

## 3. Conclusions

We developed a facile synthesis for the preparation of bimetallic Pd–Pt NDs with well-developed branches. The morphology and size of the Pd–Pt NDs were highly sensitive to the concentration of CTAC, which was used as a surfactant. The Pd–Pt NDs exhibited excellent electrocatalytic performances for both HER and MOR compared with both commercial Pd/C and Pt/C catalysts. The investigations, through electrochemical reactions, have established that the Pd–Pt NDs have superb activity and stability for both HER and MOR compared with both commercial Pd/C and Pt/C catalysts. This demonstrates that the co-influence of dendritic shape and bimetallic Pd–Pt nature is the pivotal factor for their enhancement in electrocatalytic performance. In particular, the Pd–Pt–200$_{CTAC}$ NDs showed superior electrocatalytic performances compared to their Pd–Pt ND counterparts, which could be due to the numerous unsaturated surface atoms in unique well-developed branches.

## 4. Materials and Methods

### 4.1. Materials

CTAC (Aldrich, 25 wt. %, St. Louis, MO, USA), $K_2PtCl_4$ (Aldrich, 98%), $Na_2PdCl_4$ (Aldrich, 98%), AA (Dae Jung Chemicals & Metals Co., 99.5%, Siheung, Korea), Pd/C (Alfa Aesar, 20 wt. %, Ward Hill, MA, USA), Pt/C (Alfa Aesar, 20 wt. %), potassium hydroxide (Dae Jung Chemicals & Metals Co., 93%), and methanol (Junsei, 99.8%, Tokyo, Japan) were used as received. Deionized (DI) water was used in the preparation of chemical solutions.

### 4.2. Synthesis of Pd–Pt NDs

For synthesis of Pd–Pt–200$_{CTAC}$, Pd–Pt–100$_{CTAC}$, Pd–Pt–50$_{CTAC}$, and Pd–Pt–10$_{CTAC}$ NDs, aqueous solutions of CTAC (1 mL, X mM; X = 10, 50, 100, and 200 mM) were added into reaction mixtures containing $Na_2PdCl_4$ (0.5 mL, 5 mM), $K_2PtCl_4$ (0.5 mL, 5 mM), AA (0.5 mL, 100 mM), and DI water (46 mL) with vigorous stirring for 5 min, respectively. Subsequently, the reaction mixture was heated at 90 °C for 10 min. The resultant reaction

mixture was subjected to centrifugation and washed with ethanol/DI water (8000 rpm for 5 min, 3 times) to remove excess chemicals.

### 4.3. Characterization

Transmission electron microscopy (TEM, Tokyo, Japan) and scanning electron microscopy (SEM, Tokyo, Japan) images of the prepared Pd–Pt NDs were obtained on Jeol JEM-2100F and Jeol JEM-7210F, respectively. Inductively coupled plasma optical emission spectrometry (ICP-OES, Illinois, USA) measurement was carried out using a Spectroblue-ICP-OES (Ametek). X-ray diffraction (XRD) measurements (Tokyo, Japan) were conducted on a Rigaku D/MAX2500V/PC for $2\theta$ at 20 to 90 degrees. For preparation of TEM sample, first, residual chemicals in the Pd–Pt ND solution were removed by centrifugation. The precipitated Pd–Pt NDs were re-dispersed into purified water and then an aqueous solution including nanoparticles was dropped on TEM grid (Formvar/Carbon 300 Mesh, Copper, MA, USA). Subsequently, the TEM grid was dried at room temperature. Before taking TEM image, the TEM grid was cleaned with ethanol.

### 4.4. Electrochemical Performance

Electrochemical experiments were conducted in a three-electrode cell using EC-Lab Biologic Model SP–300 potentiostat. Pt wire was use as counter electrode. Ag/AgCl (3.0 M KCl) electrode for HER and Hg/HgO (in 3.0 M NaOH) electrode for MOR were used as the reference electrodes, respectively. For the MOR electrocatalytic measurement, $N_2$-purged electrolytes (1.0 M KOH) were used for MOR. Before electrochemical experiments, catalyst ink (5 µL) was loaded onto the GCE (0.196 cm$^{-2}$).

HER electrocatalytic measurements were performed at room temperature. The $N_2$-purged electrolyte (0.5 M $H_2SO_4$) was used for HER. Before electrochemical experiments, catalyst ink (5 µL) was loaded onto the L-type GCE (0.196 cm$^{-2}$). The potential applied for linear sweep voltammogram (LSV) was between 0.05 V and −0.25 V vs. RHE with sweep rate of 10 mV s$^{-1}$.

**Author Contributions:** Conceptualization, R.K.P. and J.W.H., formal analysis. Y.W.L.; data curation J.-H.S.; writing—original draft preparation R.K.P.; writing—review and editing J.W.H. and Y.W.L. All authors have read and agreed to the published version of the manuscript.

**Funding:** This research received no external funding.

**Acknowledgments:** This work was supported by the 2021 Research Fund of University of Ulsan.

**Conflicts of Interest:** The authors declare no conflict of interest.

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
