# Peer review of "Highly Enhanced Electrocatalytic Performances with Dendritic Bimetallic Palladium-Based Nanocrystals"

_catalysts, doi:10.3390/catal11111337_

Round 1

Reviewer 1 Report

The article is devoted to the synthesis of a new form of Pt-Pd catalyst for electrochemical processes in fuel cells. The authors presented interesting results that can be used to prepare commercial samples. However, I have a number of questions and comments.

  1. The authors compare the activity of their Pt-Pd catalysts with those of commercial Pt/C and Pd/C samples. This is correct, however, for greater accuracy, you should compare your data with known Pt-Pd samples that are described in the literature.
  2. The authors believe that the high activity of the proposed catalysts is associated with their special geometric shape. However, the catalytic properties may also be due to the electronic state of the surface of metallic dendrites.
  3. It would be very interesting to see the XPS spectra of the surface of the studied samples. Do metals affect each other's electronic state?

The article can be published after the revision.

Author Response

We deeply appreciate the referee for giving favorable and valuable comments. Consulting the referee’s comments, we rigorously revised our previous manuscript. Followings are our point-by-point response to the referee’s comments. We have noted a copy of the manuscript to show where the changes have been made.

  1. Comment:The authors compare the activity of their Pt-Pd catalysts with those of commercial Pt/C and Pd/C samples. This is correct, however, for greater accuracy, you should compare your data with known Pt-Pd samples that are described in the literature.”

Our response:

We thank the reviewer for this suggestion. The electrocatalytic performance of our Pd–Pt–200CTAC NDs was compared with previously reported bimetallic Pd-Pt based catalysts, and we added related sentences and text with a comparison table in the revised manuscript as follow:

----------------------------------------------------------------------------------------------------------------

Newly added Table:

[Table 2. Comparative results for electrochemical MOR activity based on previous reports using Pd–Pt based catalysts in the literature.]

Catalyst

Electrochemical Condition

Mass Activity

(A mg-1total)

Ref.

Pd-Pt-200CTAC Nanodendrites

1 M KOH + 1 M Methanol

6.40

In this work

Pd45Pt55 Nanowires

1 M KOH + 1 M Methanol

~1.90

[34]

h-BN/PdPt Nanocorals

1 M KOH + 0.5 M Methanol

0.96

[35]

PtPd/RGO nanogarlands

1 M KOH + 1 M Methanol

0.33

[36]

Pt50Pd50 Nanocubes

1 M KOH + 1 M Methanol

0.34

[37]

PdPt/CNTs

0.5 M KOH + 0.5 M Methanol

1.07

[38]

Pt30Pd70/C

1 M KOH + 1 M Methanol

0.72

[39]

o-PdH0.43@Pt Nanooctahedra

1 M KOH + 1 M Methanol

3.68

[27]

c-PdH0.43@Pt Nanocubes

1 M KOH + 1 M Methanol

2.14

[27]

PtPd Nanowires

1 M KOH + 1M Methanol

4.29

[40]

PdPt Bimetallic Nanosponges

1 M KOH + 1 M Methanol

~2.20

[41]

Pt@Pd/RGO

1 M KOH + 1 M Methanol

~0.65

[42]

References.

[1] Zhu, C., Guo, S., & Dong, S. 2012. Advanced materials24(17), 2326-2331.

[2] Zhang, H., Xu, L., Tian, Y., Liu, X., & Chen, F. 2019. ACS omega4(6), 11163-11172.

[3] Li, S. S., Zheng, J. N., Ma, X., Hu, Y. Y., Wang, A. J., Chen, J. R., & Feng, J. J. 2014. Nanoscale6(11), 5708-5713.

[4] Zhou, L. N., Wang, Z. H., Guo, S., & Li, Y. J. 2016. Chemical communications52(86), 12737-12740.

[5] Yang, G., Zhou, Y., Pan, H. B., Zhu, C., Wai, C. M., & Lin, Y. 2016. Ultrasonics sonochemistry28, 192-198.

[6] De la Cruz-Cruz, J. J., Cayetano-Castro, N., & Dorantes-Rosales, H. J. 2020. International Journal of Hydrogen Energy45(7), 4570-4586.

[7] Liu, G., Zhou, W., Ji, Y., Chen, B., & Zhang, H. 2021. Journal of the American Chemical Society143(29), 11262-11270.

[8] Zhai, Y., Zhu, Z., Lu, X., & Zhou, H. S. 2017. ACS Applied Energy Materials1(1), 32-37.

[9] Zhu, C., Guo, S., & Dong, S. 2013. Chemistry–A European Journal19(3), 1104-1111.

[10] Feng, J. X., Zhang, Q. L., Wang, A. J., Wei, J., Chen, J. R., & Feng, J. J. 2014. Electrochimica Acta142, 343-350.

----------------------------------------------------------------------------------------------------------------

----------------------------------------------------------------------------------------------------------------

Newly added sentences:

“The MOR activity of the Pd–Pt–200CTAC NDs was compared with those of previously reported Pd–Pt–based catalysts (Table S1). The Pd–Pt–200 CTAC NDs showed superior activities than the previously reported catalysts, which can be due to dendritic shape and bimetallic Pd–Pt composition structure.”

----------------------------------------------------------------------------------------------------------------

  1. Comment:The authors believe that the high activity of the proposed catalysts is associated with their special geometric shape. However, the catalytic properties may also be due to the electronic state of the surface of metallic dendrites.”

Our response:

Thank you for your constructive comment. As mentioned by the reviewer, the changed electronic state of Pd–Pt NDs can contribute to enhanced electrocatalytic performances as well as unique dendritic shapes. We confirmed the formation of the Pd-Pt core-shell structure through elemental mapping images and line profiles. It is expected that the electron density around Pt atom is increased by electron transfer from Pd atom in the Pd-Pt NDs due to higher electronegativity of Pt than Pd. This leads to a downshift of the d-band of Pt, which can weaken the binding strength between surface Pt atoms and poisoning intermediates such as CO (ref: Zhang, H. & Chen, F. 2019. ACS omega, 4(6), 11163-11172.; Kim, Y., & Han, S. W. 2016. CrystEngComm, 18(13), 2356-2362.). The enhanced CO removal capability of Pd-Pt NDs can increase of the MOR rate. Taken together, we believe that a large surface area and numerous unsaturated surface atoms by dendritic morphology and Pd-Pt bimetallic compositional structure can enhance electrocatalytic activity. Regarding to this, we revised related sentences to more clearly describe for benefit of Pd-Pt bimetallic feature as follows.

----------------------------------------------------------------------------------------------------------------

Revised R&D part:

“In addition, compared with pure Pd and Pt catalysts, Pd–Pt bimetallic catalysts could reduce the adsorption of surface poisoning intermediates due to changed electronic structures of surface atoms, boosting the electrochemical reactions.”

----------------------------------------------------------------------------------------------------------------

  1. Comments: “It would be very interesting to see the XPS spectra of the surface of the studied samples. Do metals affect each other's electronic state?”

Our response:

In many cases of Pd-Pt catalysts previously reported, the binding energy of Pt was decreased, while an increase of Pd binding energy was observed compared to pure Pt and Pd catalysts. (ref: 1) Lee, Y. W., & Han, S. W. 2017. ACS applied materials & interfaces9(50), 44018-44026; 2) Huang, D. B., & Zhou, Z. Y. 2014. Chemical Communications50(88), 13551-13554; 3) Yousaf, A. & Manzoor, S. 2017. The Journal of Physical Chemistry C121(4), 2069-2079.) This signifies that the electron of Pd atoms is transferred to Pt atoms due to the higher electronegativity of Pt (2.28) compared to that of Pd (2.20). In contrast, the electronic states of Pd and Pt in the nanostructures were not changed in previous reports; they showed a metallic state. Although measuring the electronic states of Pd and Pt in the Pd–Pt NDs using XPS is an effective method, based on previously reported findings (ref: Lee, Y. W., & Han, S. W. 2017. ACS applied materials & interfaces9(50), 44018-44026.), we believe that our Pd–Pt NDs also possess metallic state because the Pd–Pt NDs was produced under similar synthesis condition of previous reported Pd-Pt-Cu NDs with the metallic state.    

We deeply appreciate the referee once again for giving very critical and constructive comment. Thank you very much.

Reviewer 2 Report

In this manuscript, Pramadewandaru et al. present their results demonstrating improved electrocatalytic performance through the fabrication of Pd-Pt nanocrystals with a novel dendritic morphology. Their key finding is that they can control the dendritic morphology and composition of these NCs through the use of CTA and AA, and they characterize these effects in the framework of the hydrogen evolution and methane oxidation reactions, concluding that they compare favorably to commercial solutions. To explain the improved performance, they present high resolution TEM images, EDS chemical mapping, and STEM imaging from a range of particle sizes.

Overall, I feel that this work shows nice progression in this field and I feel that it is worth publishing. Having said that, reading it raised a few questions in my mind, and I think that the authors could improve the quality of their paper if they were to address them.

The first and most important point is the authors’ claim that the evidence presented in figure 1b supports their interpretation that the enhanced performance derives from active surface metal atoms. While I don’t dispute this claim – previous works (also cited by the authors) have drawn this conclusion themselves – I do question whether the insert image in figure 1b actually shows this.  This is a phase contrast HRTEM image, and, as such, it can be very difficult to interpret. In particular, the ND is sitting atop an amorphous carbon support film, and the contrast highlighted by the yellow arrow actually looks more like it arises from this rather than a surface atom. To convince me, the authors would need to include a detailed computer simulation of the contrast under the experimental conditions, which is sufficiently complex for this system to be a paper in its own right!  I therefore suggest that the authors simply remove this claim.  I don’t feel it detracts from their overall findings, but it does mean that they will need to rephrase their discussion to be more cautious.  I would also encourage them to investigate this more explicitly in a future publication.

The second point is the EDS maps.  I do agree that they probably show a core-shell structure; however, they are really too sparse to be convincing.  The authors could try blurring them a bit to improve their perceptive quality, especially in a publication.  Most commercial EDS software offers this option. This relatively wide field of view does not require high resolution.

Third, how were the error bars on the size distributions calculated?  This should be described in detail.  I presume that they come from image analysis of the TEM / STEM data?  If so, how were the images segmented?  This is not trivial for agglomerates with overlapping particles (found in many of the images here).

Fourth, a description of the TEM sample preparation should be included for good measure.  The sample preparation admittedly looks excellent, so this would be a very useful point to add.

Finally, please check the numbering of the sections.  Alternatively, consider placing the methods section directly after the introduction (or possibly after the conclusions).  Based on numbering errors observed in section 3, it seems that this was the original plan, and I think the narrative would flow better this way (it was a bit jarring to be confronted with the methods section when I was just getting ready to read the conclusions).

Author Response

Response to the Reviewer 2’s Comments

We deeply appreciate the referee for giving favorable and valuable comments. Consulting the referee’s comments, we rigorously revised our previous manuscript. Followings are our point-by-point response to the referee’s comments. We have noted a copy of the manuscript to show where the changes have been made.

  1. Comments:I do question whether the insert image in figure 1b actually shows this. This is a phase contrast HRTEM image, and, as such, it can be very difficult to interpret. In particular, the ND is sitting atop an amorphous carbon support film, and the contrast highlighted by the yellow arrow actually looks more like it arises from this rather than a surface atom. To convince me, the authors would need to include a detailed computer simulation of the contrast under the experimental conditions, which is sufficiently complex for this system to be a paper in its own right! I therefore suggest that the authors simply remove this claim. I don’t feel it detracts from their overall findings, but it does mean that they will need to rephrase their discussion to be more cautious. I would also encourage them to investigate this more explicitly in a future publication.”

Our response:

Thank you for your critical comment. To clearly show the unsaturated atoms on the surface of Pd–Pt NDs, we newly obtained TEM images of the Pd–Pt NDs and corresponding surface atom model was constructed (Figure R1). The newly obtained TEM image and model clearly show the existence of low-coordinated atoms on the Pd–Pt NDs. In the previous publications (ref:  Zhang, L., & Sun, X. 2019. Energy & Environmental Science12(2), 492-517; Li, T., & Wang, F. 2018. ACS Catalysis8(9), 8450-8458.), low-coordinated surface atoms of the Pd-Pt NDs showed much higher electrocatalytic activities for methanol oxidation and superior CO tolerance compared to commercial Pt/C and Pd/C. In addition, low-coordinated surface atoms can be active sites for HER and other electrocatalysis (ref: Cheng, N., & Sun, X. 2016. Nature communications7(1), 1-9.). The newly obtained TEM images were added in Figure 1b and inset as follows.

-------------------------------------------------------------------------------------------------------------

Revised Figure:

[Figure R1. HRTEM image of Pd-Pt NDs and corresponding surface atom model.]

-------------------------------------------------------------------------------------------------------------

Revised Figure:

[Figure 1. (a) and (b) HR–TEM image including the correspond FFT images. HAADF-STEM image with (c) corresponding EDS elemental mapping images, and (d) line-scan profiles analysis of Pd (black) and Pt (red) for the Pd–Pt NDs. (e) XRD pattern confirm of corroborating the Pd-Pt bimetallic feature.]

-------------------------------------------------------------------------------------------------------------

  1. Comments:The second point is the EDS maps. I do agree that they probably show a core-shell structure; however, they are really too sparse to be convincing. The authors could try blurring them a bit to improve their perceptive quality, especially in a publication. Most commercial EDS software offers this option. This relatively wide field of view does not require high resolution.”

Our response:

Thank you for your critical comment. As mentioned by the reviewer, elemental mapping is somewhat unclear to determine the compositional structure of the Pd–Pt NDs as core-shell structures. To confirm the compositional structure of the Pd–Pt NDs, a compositional line profile was obtained. In the line profile of the Pd–Pt NDs, only Pd signal was observed at the outer part of the Pd–Pt NDs and both Pd and Pt signals were detected in the middle part of the Pd–Pt NDs. This signifies the formation of the Pd-Pt core-shell structure. In the revised manuscript, we added a STEM image of a Pd–Pt ND used for elemental line profile in the inset of Figure 1d as follows.   

-------------------------------------------------------------------------------------------------------------

Revised Figure:

[Figure 1. (a) and (b) HR–TEM image including the correspond FFT images. HAADF-STEM image with (c) corresponding EDS elemental mapping images, and (d) line-scan profiles analysis of Pd (black) and Pt (red) for the Pd–Pt NDs. (e) XRD pattern confirm of corroborating the Pd-Pt bimetallic feature.]

-------------------------------------------------------------------------------------------------------------

  1. Comments:Third, how were the error bars on the size distributions calculated? This should be described in detail. I presume that they come from image analysis of the TEM / STEM data? If so, how were the images segmented? This is not trivial for agglomerates with overlapping particles (found in many of the images here).”

Our response:

Thank you for your critical comment. We obtained the particle size distributions of nanostructures using imageJ (software). In this work, we used more than 200 nanostructures for determining the size distributions. To clearly show the size distribution of our nanostructures, size distribution histogram was added in of Figure 2i-l as follow.

-------------------------------------------------------------------------------------------------------------

Revised Figure:

[Figure 2. SEM, TEM images and histogram of the particles size distribution of Pd-Pt NDs with controlled CTAC concentration representative to (a); (e); (i) Pd–Pt–10CTAC NDs, (b); (f); (j) Pd–Pt–50CTAC NDs, (c); (g); (k) Pd–Pt–100CTAC NDs, and (d); (h); (l) Pd–Pt–200CTAC NDs.]

  1. Comments:Fourth, a description of the TEM sample preparation should be included for good measure. The sample preparation admittedly looks excellent, so this would be a very useful point to add.”

Our response:

In typical TEM sample preparation, we generally used a conventional preparation method for nanoparticles. First, after removing excess chemicals by centrifugation, the precipitates (nanoparticles) were re-dispersed into purified water, and then the aqueous solution including nanoparticles was dropped on the TEM grid (Formvar/Carbon 300 Mesh, Copper). Subsequently, the TEM grid was dried at room temperature. Before taking the TEM image, the TEM grid was cleaned with ethanol. This was added in the revised manuscript as follows.

----------------------------------------------------------------------------------------------------------------

Newly added sentences:

“For preparation of TEM sample, first, residual chemicals in the Pd-Pt ND solution were removed by centrifugation. The precipitated Pd-Pt NDs were re-dispersed into purified water and then an aqueous solution including nanoparticles was dropped on TEM grid (Formvar/Carbon 300 Mesh, Copper). Subsequently, the TEM grid was dried in room temperature. Before taking TEM image, the TEM grid was cleaned with ethanol.”

----------------------------------------------------------------------------------------------------------------

  1. Comments:Finally, please check the numbering of the sections. Alternatively, consider placing the methods section directly after the introduction (or possibly after the conclusions). Based on numbering errors observed in section 3, it seems that this was the original plan, and I think the narrative would flow better this way (it was a bit jarring to be confronted with the methods section when I was just getting ready to read the conclusions).”

Our response:

Thank you for your constructive comment. As suggested by the reviewer, we arranged ‘Materials and Methods’ part in the end of manuscript in the revised manuscript as follow. 

----------------------------------------------------------------------------------------------------------------

Revised Experimental part:

  1. Materials and Methods

4.1. Materials

CTAC (Aldrich, 25 wt%), K2PtCl4 (Aldrich, 98%), Na2PdCl4 (Aldrich, 98%), AA (Dae Jung Chemicals & Metals Co., 99.5%), Pd/C (Alfa Aesar, 20wt%), Pt/C (Alfa Aesar, 20wt%), potassium hydroxide (Dae Jung Chemicals & Metals Co., 93 %), and methanol (Junsei, 99.8%) were used as received. Deionized (DI) water was used in the preparation of chemical solutions.

4.2. Synthesis of Pd–Pt NDs

For synthesis of Pd–Pt–200CTAC, Pd–Pt–100CTAC, Pd–Pt–50CTAC, and Pd–Pt–10CTAC NDs, an aqueous solutions of CTAC (1 mL, X mM; X= 10, 50, 100, and 200 mM) were added into reaction mixtures containing of Na2PdCl4 (0.5 mL, 5 mM), K2PtCl4 (0.5 mL, 5 mM), AA (0.5 mL, 100 mM), and DI water (46 mL) with a vigorous stirring for 5 min, respectively. Subsequently, the reaction mixture was heated at 90 °C for 10 min. The resultant reaction mixture was subjected to centrifugation and washed with ethanol/DI water (8000 rpm for 5 min, 3 times) to remove excess chemicals.

4.3. Characterization

Transmission electron microscopy (TEM) and scanning electron microscopy (SEM) images of the prepared Pd–Pt NDs were obtained on Jeol JEM-2100F and Jeol JEM-7210F, respectively. Inductively coupled plasma-optical emission spectrometry (ICP-OES) measurement was carried out using a Spectroblue-ICP-OES (Ametek). X-ray diffraction (XRD) measurement were conducted on a Rigaku D/MAX2500V/PC scanning for 2Ó¨ at 20 to 90 degrees. For preparation of TEM sample, first, residual chemicals in the Pd-Pt ND solution were removed by centrifugation. The precipitated Pd-Pt NDs were re-dispersed into purified water and then an aqueous solution including nanoparticles was dropped on TEM grid (Formvar/Carbon 300 Mesh, Copper). Subsequently, the TEM grid was dried in room temperature. Before taking TEM image, the TEM grid was cleaned with ethanol.

4.4. Electrochemical performance

Electrochemical experiments were conducted in a three-electrode cell using EC-Lab Biologic Model SP–300 potentiostat. Pt wire was use as counter electrode. Ag/AgCl (3.0 M KCl) electrode for HER and Hg/HgO (in 3.0 M NaOH) electrode for MOR were used as the reference electrodes, respectively. For the MOR electrocatalytic measurement, N2-purged electrolyte (1.0 M KOH) were used for MOR. Before electro-chemical experiments, catalyst ink (5 μL) was loaded on the GCE (0.196 cm–2).

HER electrocatalytic measurements were performed at room temperature. The N2-purged electrolyte (0.5 M H2SO4) was used for HER. Before electrochemical experi-ments, catalyst ink (5 μL) was loaded on the L-type GCE (0.196 cm–2). The potential applied for linear sweep voltammogram (LSV) between 0.05 V to -0.25 V vs. RHE with sweep rate of 10 mV s–1.

----------------------------------------------------------------------------------------------------------------

Finally, we deeply appreciate the referee once again for giving very critical and constructive comment. Thank you very much.

Round 2

Reviewer 1 Report

I am satisfied with the revised version of the article and recommend its publication in current form.